# Supercritical Fluid Technologies for the Incorporation of Synthetic and Natural Active Compounds into Materials for Drug Formulation and Delivery

**DOI:** 10.3390/pharmaceutics14081670

**Published:** 2022-08-11

**Authors:** Katja Andrina Kravanja, Matjaž Finšgar, Željko Knez, Maša Knez Marevci

**Affiliations:** Faculty of Chemistry and Chemical Engineering, University of Maribor, Smetanova 17, 2000 Maribor, Slovenia

**Keywords:** supercritical fluid technologies, active compounds, drug formulation, drug delivery

## Abstract

Various active compounds isolated from natural sources exhibit remarkable benefits, making them attractive for pharmaceutical and biomedical applications, such as antioxidant, antimicrobial, and anti-inflammatory activities, which contribute to the treatment of cardiovascular diseases, neurodegenerative disorders, various types of cancer, diabetes, and obesity. However, their major drawbacks are their reactivity, instability, relatively poor water solubility, and consequently low bioavailability. Synthetic drugs often face similar challenges associated with inadequate solubility or burst release in gastrointestinal media, despite being otherwise a safe and effective option for the treatment of numerous diseases. Therefore, drug-eluting pharmaceutical formulations have been of great importance over the years in efforts to improve the bioavailability of active compounds by increasing their solubility and achieving their controlled release in body media. This review highlights the success of the fabrication of micro- and nanoformulations using environmentally friendly supercritical fluid technologies for the processing and incorporation of active compounds. Several novel approaches, namely micronization to produce micro- and nano-sized particles, supercritical drying to produce aerogels, supercritical foaming, and supercritical solvent impregnation, are described in detail, along with the currently available drug delivery data for these formulations.

## 1. Introduction

Despite considerable success in developing new drugs, trends show that the biopharmaceutical potential of many newly manufactured chemical products has not yet been realized because they suffer from poor solubility in aqueous media, low permeability, and they are rapidly metabolized and excreted from the body with low tolerability due to the increasing development of drugs with greater lipophilicity and higher molecular weight. Poor solubility is the most challenging aspect and represents the majority of failures in the development of new pharmaceuticals, accounting for approximately 40% of drugs with marketing approval and 90% in the discovery pipeline. Such drugs with poor solubility are classified as Class 2 and Class 4 in the Biopharmaceutical Classification System (BCS), which necessitates the exploration of drug formulations using micro- and nano-encapsulation or micronization techniques to improve the in vitro and in vivo performances of such drug candidates and, consequently, their bioavailability [1,2].

Proteins, polysaccharides, vitamins, minerals, antioxidants, and enzymes from natural sources are also promising candidates for use in the nutraceutical, pharmaceutical, biomedical, and cosmetic industries due to their diverse benefits, abundance in nature, and affordability [3,4]. Proven positive therapeutic effects and growing environmental concerns are the main motivations for researching natural bioactive compounds as possible alternatives to synthetic drugs [5]. Growing evidence suggests that plant polyphenols, which include anthocyanidins, catechins, flavanones, flavones, flavonols, isoflavones, hydroxybenzoic acids, hydroxycinnamic acids, lignans, and tannins, play an important role in the prevention of numerous diseases [4]. Polyphenols are usually secondary metabolites of plants, consisting of an aromatic ring with one or more hydroxyl groups. They act as antioxidants [6], have antibacterial and antifungal effects [7,8], and may alter the expression of genes in the inflammatory pathway [9,10], thus playing a protective role against cancer [11], cardiovascular disease, diabetes, and Alzheimer’s disease [12,13]. However, their disadvantages lie mainly in their instability and insolubility in body fluids and challenging adsorption through membranes, resulting in poor bioavailability in the body [4,14].

For synthetic and natural active ingredients (AIs), current research is focused on eliminating environmentally harmful chemicals and processing methods by implementing environmentally friendly technologies [5]. The formulation of AIs with supercritical fluids (SCFs) is one of the leading strategies incorporating environmentally friendly and economically promising characteristics, and it provides quality products without using high temperatures that could lead to the thermal degradation of AIs [15,16,17]. The higher bioavailability of AIs is feasible by means of supercritical (SC) micronization [18,19] or by the SCF-guided encapsulation of AIs into various polymeric matrices or frameworks [15,17,20].

This review comprehensively describes various well-established SCF technologies for the formulation of AIs. It includes micronization methods, namely the rapid expansion of supercritical solutions (RESS), antisolvent methods, and particles from gas-saturated solutions (PGSS^TM^), which can improve bioavailability by reducing particle size, and the technologies for encapsulating AIs in polymer carriers. The review also addresses the preparation of aerogels and foams as porous drug carrier formulations for AIs. Finally, SC solvent impregnation is used to incorporate AIs into prepared polymer carriers. Depending on the role of the SCF as a solvent, antisolvent, or solute, the choice of the carrier material, SCF interactions with the AIs and polymeric carriers, and the selected operating conditions, different products with characteristic drug release kinetics can be fabricated. Therefore, this paper will discuss in detail drug delivery from prepared SCF formulations and highlight the future prospects in this field.

## 2. SCF Technologies for the Incorporation of AIs

In the last few decades, tremendous progress has been made in developing SCF technologies for drug formulation. The main reasons for their use in the pharmaceutical field are environmental concerns, as the processes are carried out in the absence of organic solvents with detrimental effects, the production of high-quality products by utilizing SCFs with adjustable properties under different processing conditions, and cost efficiency [21]. A wide selection of substances can be used as SCFs, which can exist as a single phase above the critical conditions, such as H_2_O, N_2_, Xe, SF_6_, N_2_O, C_2_H_4_, CHF_3_, ethylene, propylene, propane, ammonia, n-pentane, ethanol, acetone, etc. However, CO_2_ has most frequently been implemented for numerous reasons (Figure 1) [22]. It is an inert, nonpolar, non-inflammable, and inexpensive gas generally recognized as safe (GRAS) by the Food and Drug Administration (FDA) [23,24]. The main advantage of supercritical CO_2_ (SC-CO_2_) is its low critical point at *T* = 304.21 K and *P* = 7.382 MPa, indicating low operating costs and convenience when working with thermosensitive pharmaceutical compounds [25]. In addition, CO_2_ has excellent transport properties in the SC state due to its liquid-like density, gas-like diffusivity, and viscosity between gas and liquid [26], which can be tunable with varying process parameters. It provides suitable solubility for polymers, while various co-solvents can be added to increase the miscibility of polar substances. It is recyclable and leaves little to no trace in the final product, as it can be easily separated in the final stage of the process by depressurization to its gaseous state [25,27].

The selection of the supercritical process for the encapsulation of AIs is based mainly on the product’s desired morphology, the solvents’ thermodynamic properties, the encapsulation materials and AIs used, and the solubility of the AI in SC-CO_2_. Different AI formulations can be obtained depending on the role played by SC-CO_2_ as a solvent, antisolvent, solute, drying medium, or foaming agent, which are presented in detail below [22,24].

### 2.1. Micronization

Micronization is a size reduction technique that produces small particles of less than 10 µm, which significantly increases the bioavailability of AIs with poor water solubility due to the improved dissolution rate of micro- and nanosized compounds in gastrointestinal media. When smaller particles are used, their solubility increases due to their higher surface area in contact with water, crystal lattice defects, and changes in surface thermodynamic properties, therefore requiring lower drug dosages [28,29]. The choice of micronization technology greatly influences the shape, agglomeration behavior, particle size, and size distribution [30]. In addition to SCF technologies, mechanical comminution, spray drying, and other conventional methods are used in this regard but are associated with the disadvantages of having a broader particle size distribution, and often utilizing high temperatures and organic solvents [29]. SCF micronization processes can be classified into three categories depending on the SC-CO_2_ role, which can serve as a solvent by using the rapid expansion of supercritical solutions (RESS) and derived processes, as an antisolvent with the use of a supercritical antisolvent (SAS) and derived processes, or as a solute by using particles from gas-saturated solutions (PGSS^TM^) and derived processes (Figure 2) [31].

#### 2.1.1. RESS

RESS is a micronization process first reported in the late 1980s [35] in which SC-CO_2_ plays the role of a solvent. The process consists of two sequential steps. The first step includes the AI being dissolved in SC-CO_2_ (alone or in combination with the coating material, e.g., a polymer), followed by a second step of rapid depressurization (<10^−6^ s) through a nozzle, which leads to a density drop and high supersaturations. The consequent high nucleation rate limits crystal growth and allows the formation of particles smaller than 500 nm with a narrow size distribution and high purity. The apparatus is comprised of four main units, namely a dissolution unit where solids dissolve in SC-CO_2_, a thermostatted pre-expansion unit, a nozzle through which the SC solution is expanded to ambient temperature (*T*) and pressure (*P*), and a post-expansion unit where the gas is separated from the microparticles [32].

The final particle morphology obtained by the RESS process is influenced by several parameters, such as *T*, *P*, the selection of a co-solvent, the mass flow rate, and the AI concentration in SC-CO_2_ [32]. In addition, these parameters significantly affect the release of processed drugs in simulated body fluids (SBF). In particular, the choice of encapsulation material and its ratio to the drug was found to have a greater influence on the drug dissolution rate than the influence of particle size [36,37,38]. Vergara-Mendoza et al. [38] showed that the coenzyme Q_10_ (coQ_10_) was encapsulated in either poly(ethylene glycol) (PEG) or poly(lactic acid) (PLA), which were dissolved in SC-CO_2_ and co-solvent ethanol or acetone. Figure 3 shows a comparison of the drug dissolution data obtained by different processing parameters, such as the selected encapsulating polymer, the polymer/AI ratio, and the selected co-solvent. The drug dissolved better when ethanol was used as the co-solvent. Interestingly, drug dissolution increases at higher polymer concentrations, i.e., with a ratio of 1/0.5 or 2/1, and decreases at a polymer to AI ratio of 1/1. This phenomenon occurs regardless of the final particle size, as a higher release of coQ_10_ can be observed for PLA when the polymer concentration increases despite a larger particle diameter [38].

Despite the production of very small particles with a narrow size distribution, the main drawbacks of RESS are related to the high gas demand due to the low solubility of the solids (the ratio of SCF to solutes ranges from 10 to 1000 kg/kg). In addition, the solubility of the solutes increases with pressure; therefore, the process generally operates at pressures higher than 10 MPa. It is estimated that the process parameters of RESS typically range between 10–40 MPa and 308–333 K, resulting in relatively high operating costs. Together with the difficulty of separating the fine particles from the gas volume, these drawbacks have limited the application of the RESS process to small scale applications [27,39].

#### 2.1.2. Antisolvent Processes

When an AI is poorly soluble or insoluble in SC-CO_2_ (including gaseous or liquid CO_2_), the latter can be used as an antisolvent to obtain micronized AI particles. The first antisolvent process was introduced in 1989 and is referred to as the gas antisolvent process (GAS). It involves an AI (or an AI in combination with the carrier) being dissolved in a liquid organic solvent that is readily miscible with CO_2_, followed by the solution being brought into contact with CO_2_, preferably injected into the precipitation unit from the bottom. CO_2_ dissolves in the organic solvent and reduces its solvent power, resulting in the supersaturation and precipitation of the solutes, which are collected after depressurization. The supersaturation leads to a high nucleation rate and the formation of smaller particles [23,40].

Several variants of antisolvent processes differ in the initial feedstock and as to how different phases are brought into contact. For example, GAS and the supercritical fluid extraction of emulsions (SFEE) differ regarding feedstock, as SFEE uses emulsions as starting materials, whereby SC-CO_2_ is used to extract the organic phase of the emulsion. However, while GAS operates without a nozzle, the supercritical antisolvent process (SAS) uses a nozzle to atomize a mixture of the solvent and the AI. It works on the principle that droplets of the organic solvent and the AI are sprayed into the precipitation unit that is already filled with CO_2_ in the supercritical state, causing the AI to precipitate [40]. Over the years, many modifications of SAS have been developed, such as solution-enhanced dispersion by supercritical fluids (SEDS), the atomization of a supercritical antisolvent-induced suspension (ASAIS), the atomization and antisolvent precipitation process (AAS), the aerosol solvent extraction system (ASES), SAS with enhanced mass transfer (SAS-EM), etc. This review focuses on SAS, the most commonly used antisolvent process, since detailed descriptions of the variants can be found elsewhere [23,41,42,43] and are beyond the scope of this review [23,42].

Unlike GAS, SAS can operate continuously and is therefore suitable for industrial scale application [40]. However, a deeper understanding of the process is still required for more common use in the pharmaceutical industry, as particles with an irregular shape, broad particle size distribution, and low encapsulation efficiency have previously been collected. Since the process involves three components, the ternary phase equilibrium must be carefully observed to obtain particles with the desired properties [23]. Both crystalline and amorphous products can be obtained by adjusting the operating conditions. Crystalline materials are precipitated at fixed temperatures and a *P* below the critical point of the mixture. Meanwhile, amorphous materials can be obtained by increasing the *P* to values well above the critical point and have been shown to accelerate the drug dissolution rate in aqueous media [44,45].

Supercritical antisolvent fractionation (SAF) is a subvariant of the SAS method. This relatively new method is in the foreground for the purification of target compounds with high yield and purity. It exploits the likelihood that different compounds in the feed mixture (e.g., a natural extract containing various bioactive compounds) will precipitate differently under the same process conditions, such as *P*, *T*, the amount of solvent, and the choice of antisolvent [46]. When SC-CO_2_ is brought into contact with the feed mixture and organic solvent under pressure, the SC-CO_2_ dissolves the nonpolar compounds that are soluble in the fractionation media and leads to selective precipitation of the polar compounds. [47,48]. For example, Villalva et al. used SAF to fractionate *Achillea millefolium* L. extract, resulting in two fractions, one rich in phenolic compounds with high antioxidant potential, and the other rich in essential oil with high anti-inflammatory activity [49]. The complete or partial separation and enrichment of a mixture containing AIs is feasible by obtaining two or more fractions of compounds [47].

#### 2.1.3. PGSS^TM^

PGSS^TM^ uses SC-CO_2_ as a solute and is suitable for the micronization of AIs and coating materials insoluble in SC-CO_2_, but it can absorb large quantities of gas, which lowers their melting point. An autoclave is filled with substances (AI alone or in combination with a coating material) that are melted, emulsified, or suspended in the liquid. SC-CO_2_ is then introduced into the autoclave to dissolve in the melt and form a gas-saturated solution. The contents of the autoclave are then passed through a nozzle and sprayed in a spray tower at atmospheric *P*, resulting in expansion. During expansion, the mixture’s temperature drops considerably due to the Joules–Thompson effect, forming solidified micron-sized particles separated from the stream of gaseous CO_2_ in a cyclone [39,50,51].

PGSS^TM^ is one of the SC micronization processes most commonly used on an industrial scale due to several advantages. First, the process is exceptionally economical due to the low consumption of the SCF (starting at 1 g of CO2 per g of substance mixture), the high precipitation yields, and operation at moderate *T* and *P*. In addition, the process can be operated in batch mode or continuously, with relatively easy scale-up and low investment costs. The micronized particles are also solvent-free and have the narrowest size distribution, despite the disadvantage of the process being unable to fabricate submicron-sized particles [22,27,39].

Various AIs have been formulated using PGSS^TM^ [52,53,54,55,56,57,58,59,60,61,62]. Spheres, fibers, and porous particles can be produced [54]. The morphology of the products varies depending on the drug-loading level in the solid dispersion (SD), the *P* and *T* in the autoclave, the filling rate of the autoclave, the nozzle diameter, the agitation speed, and the agitation time [52,53]. The impact of these parameters on drug dissolution was evaluated using an experimental design approach in a study in which composite particles of SD-containing fenofibrate and gelucire were prepared by PGSS^TM^. The most influential parameters were the autoclave *T* and *P*, and the drug loading level in SD, and the optimal conditions found were *T* = 78 °C, *P* = 80 bar, and wt.% = 220 mg drug per g SD. [52]. The selection of the carrier materials is also critical for obtaining the desired drug kinetics. For example, S-(+)-ibuprofen was encapsulated in different carrier materials by PGSS^TM^. Drug release tests in simulated gastric and intestinal fluids demonstrated the faster solubility of ibuprofen for poloxamer as the carrier material, while sustained, controlled release was observed for gelucire and glyceryl monostearate [54].

### 2.2. SC Drying for the Preparation of Aerogels

Since Kistler’s invention thereof in 1931 [63], aerogels have been of great interest for various applications, namely as thermal insulators in construction [64], in the packaging, textile, and cosmetics industries [65], as catalysts [66], in the development of biosensors [67], as energy storage devices [68], for space applications [69], as bioactive coatings in the biomedical field [70], and as carriers of AIs in the pharmaceutical industry [71,72,73,74]. They are known for their exceptional properties, as presented below:I.a three-dimensional, highly porous structure (with a pore diameter smaller than 100 nm) [75],II.a very low density (0.0011–0.5 g/cm^3^) [75],III.a large specific surface area (70–1600 m^2^/g) [76],IV.low thermal conductivity (as low as 0.012 W/mK in air at 1 atm and 300 K) [77],V.a low dielectric constant,VI.a low speed of sound, andVII.a low refractive index [75,78].

Kistler defined aerogels as materials derived from wet gels in which the liquid in the pores is replaced by gas under conditions that preserve their volume [75]. A more exact definition also describes them as an open, solid, colloidal, or polymeric network consisting of loosely packed, interconnected particles or fibres [65].

They are produced by sol-gel synthesis and subsequent drying of the gels produced. The process begins with the formation of a colloidal suspension “sol”, which is formed by dispersing the precursor particles in the selected solvent. Adding a catalyst to the solution stimulates polymerization reactions involving hydrolysis and polycondensation. The physical or chemical crosslinking of the polymer leads to the formation of a wet gel, a two-phase system consisting of a solid three-dimensional network and a solvent entrapped in its pores. The bonds formed during gelation are initially weak. Therefore, gel aging is required to strengthen the bonds in the network and to achieve mechanical stability. Finally, the prepared wet gel is dried to remove the liquid solvent in the pores of the gel and replace it with gas. Depending on the drying method, different end products can be produced. Air drying or evaporation produces xerogels, freeze-drying/lyophilization produces cryogels, and supercritical drying produces aerogels [79,80]. In SCF drying, the transition from the liquid to the gaseous state is achieved without directly crossing the phase boundary between the liquid and gaseous state. Instead, the transition occurs through the supercritical region, which can avoid surface tension and the consequent failure of the structure. Therefore, the advantage of drying with a SCF is that it preserves the structural characteristics of the wet gel and forms highly porous end materials with a high specific surface area [75,78]. SC-CO_2_ is most commonly used for the supercritical drying of wet gels. Alcohol is often used as a substitute for the aqueous phase in the hydrogel obtained by sol-gel synthesis to prepare so-called alcogels because the alcohol in the pores is readily miscible with SC-CO_2_ and can therefore be easily removed [81,82].

Different strategies are available for the incorporation of AIs into the carrier (Figure 4). For example, AIs can be incorporated into the aerogel structure during aerogel preparation or after SC drying. Despite the need for high AI dosages, the most straightforward strategy is to add the AIs to the precursor solution of the gel before crosslinking and SC drying. However, this is only possible for AIs soluble in the precursor solution and insoluble in alcohol and the SCF to avoid premature extraction [83]. Alternatively, if the AI is soluble in alcohol and insoluble in SCF, it can be incorporated into the gel structure during the solvent exchange step in the preparation of the alcogel from the alcoholic solution [83,84]. Loading AIs that are soluble in SC-CO_2_ (e.g., various essential oils) is also feasible during SC drying [83,85,86]. After preparation, the final strategy is to load the AIs by diffusion through the pore network into the aerogels. Supercritical impregnation can achieve this, which is advantageous because only small dosages of AIs are required, there are no solvent residues, the process occurs at low working temperatures, and it is generally suitable for water-insoluble drugs [83].

Aerogels can be made from various inorganic [87,88,89] and organic materials [90,91], thus contributing to their unique characteristics and wide application. Conventional aerogels based on inorganic and petrochemical materials such as silica, graphene, titanium, and their oxides have certain shortcomings in biomedical and pharmaceutical applications. Many are not environmentally friendly; for example, silica-based aerogels are considered biocompatible but not biodegradable. Therefore, various biodegradable organic polysaccharide materials have been investigated as carriers of AIs for oral drug delivery applications because they are formulated from natural components and because they are abundant, environmentally friendly, biodegradable, and biocompatible [92]. Examples of polysaccharides used to prepare aerogels include cellulose [93], starch [94], carrageenan [73], chitosan [95], alginate [82,96], and pectin [97,98]. The structural characteristics of such aerogels differ based on their chemical properties and the methods used to prepare the gels [92,99]. Optimizing the drug release kinetics controlled by the diffusion mechanism and polymer degradation for a particular application is very important. Successful attempts have already been made to achieve the controlled release of AIs from biodegradable aerogels in simulated gastric and intestinal fluids [100,101]. In addition, the solubility of AIs with a hydrophobic character was significantly improved by their incorporation into aerogels [94,97,102].

### 2.3. SC Foaming

In SC foaming, SC-CO_2_ is used alone or in combination with other gasses as a blowing agent. The most frequently used foaming processes are batch foaming, extrusion, and injection molding. Batch foaming is mainly employed on a laboratory scale and offers good control of the processing variables. In this process (Figure 5), CO_2_ is first dissolved in a selected polymer under pressure to produce a polymer/gas solution [103]. Reducing the *P* or increasing the *T* then triggers the foaming process resulting from thermodynamic instability due to the supersaturation of the CO_2_ dissolved in the polymer [104]. The foaming process consists of bubble nucleation and bubble growth (expansion), and it is completed before the vitrification or crystallization of the polymer to avoid cell coalescence, which could lead to the rupture of the cell wall and the collapse of the foam structure.

Although the concept seems comprehensible, understanding the parameters, such as gas concentration, the diffusivity of the gas in the polymer, temperature, the decompression rate, and the thermodynamic properties of the polymer, is crucial to obtaining stabilized end products with the desired morphology. In particular, the plasticization effect of the polymer is an important parameter that greatly assists in stabilizing the foam. The glass transition temperature of the polymer is typically reduced upon exposure to CO_2_. However, when the polymer/gas solution leaves the nozzle, the CO_2_ is desorbed, resulting in the loss of the plasticization effect, which helps stabilize the foam obtained. In addition, the foam generation process requires the good solubility of the foaming agents in a selected polymer and an adequate melt strength [103]. Usually, amorphous polymers are used because they have a low glass transition temperature, such as polystyrene [106], poly(ε-caprolactone) (PCL) [107], PLA [108,109,110], poly(lactic-co-glycolic acid) (PLGA) [16,110,111,112], etc. PCL, PLA, and PLGA are used particularly in biomedical and pharmaceutical applications due to their biocompatibility, good degradability, and mechanical properties [113]. Porous matrices obtained by the SC foaming of these polymers are desirable for developing drug delivery systems as their open pore structure provides a large specific surface area allowing drug loading and controlled local release. Various strategies have been developed for incorporating drugs into foamed polymers, including single-step foaming and the drug impregnation of the polymer matrix. In this process, the polymer and drug are placed in a vessel and saturated with CO_2_, followed by establishing the supercritical conditions. When the glass temperature of the polymer falls below the process temperature, the polymer chains swell. Upon CO_2_ leaving the system by depressurization, the process of nucleation begins due to the supersaturation of the polymer matrix. During this process the active ingredient can easily disperse in the porous structure of the foam [114,115]. Such one-step systems are possible by using the polymer, SC-CO_2_ and the drug, or by adding a co-solvent to the polymer. For example, Álvarez et al. prepared a gemcitabine-loaded PLGA foam by adding the drug to an ethyl lactate and PLGA mixture, followed by the slow introduction of high-pressure CO_2_ into the system to dissolve it in the mixture [116]. Alternatively, various two-step strategies have been developed to introduce the drug into a foam. Rojas et al. [117] prepared PLA foams loaded with cinnamaldehyde by SC foaming and the subsequent CO_2_-assisted impregnation of the obtained PLA foams. Ong et al. [112] developed a two-step system consisting of an emulsification-solvent evaporation microencapsulation technique and the foaming of the polymer. Similarly, a two-step spray-drying and foaming process was presented by Lee et al. [118] and Nie et al. [111]. Current findings on drug dissolution from polymeric carriers obtained by SC foaming in most cases has revealed prolonged release in phosphate-buffered saline (PBS), often lasting more than a month, thus showing great promise for biomedical applications [110,111,118,119].

### 2.4. SC Solvent Impregnation (SSI)

SSI is used in various fields, including textile dyeing, wood impregnation, biomedicine (e.g., tissue engineering, wound dressings), and the development of controlled-release drug delivery systems [120]. The process takes advantage of the physicochemical properties of SC-CO_2_, especially its density, which is close to a liquid and ranges from 0.2 to 1.5 g/cm^3^, and its gas-like diffusivity. Due to its high density, it has good solvation power and can solubilize various compounds, while its high diffusivity enables the diffusion of hydrophobic drugs with CO_2_ into polymeric matrices. The process in the SSI apparatus (Figure 6) involves first dissolving the drug in SC-CO_2_ by adding it separately to the reactor with the polymer. For biomedical applications, impregnated polymers are typically various biodegradable polyesters (poly(l-lactic acid) (PLLA), poly(d-l-lactic acid) (P(D,L)LA), PLGA, PCL), hydrogels, aerogels, silicon-based copolymers, poly(methyl methacrylate) (PMMA), etc. The desired *P* (90–200 bar) and *T* (35–55 °C) are then established, allowing contact between SC-CO_2_ and the drug with the polymer, which facilitates the diffusion of the drug into the polymeric matrix [25]. Under these conditions, the compressed fluid may also induce swelling or act as a plasticizer agent for the polymer, thereby aiding the diffusion of the drug into the polymer matrix [121]. Afterwards, depressurization allows the CO_2_ to change into a gaseous state upon venting, and the impregnated polymer can be recovered.

Usually, the loading of the AI into the polymer is determined, and the impregnation efficiency is evaluated using the partitioning coefficient K, which indicates the relative affinity of the drug for the polymer and CO_2_ under certain conditions [25]. Depending on the contact between the AI and the polymer, impregnation can be static or dynamic. In static impregnation, the SC-CO_2_, AI, and polymer are placed in a reactor and subjected to the desired *T* and *P*, while in the dynamic method, the SC-CO_2_ is continuously passed over the polymer matrix. In both cases, the process is influenced by several parameters, namely the *P*, *T*, hydrodynamics, depressurization rate, solubility of the AI in SC-CO_2_, and diffusion coefficient [122]. When used appropriately, the physical, chemical, or mechanical properties of the AIs, polymers, and additives are not altered [121].

One of the most remarkable advantages of SSI is that no solvent residues remain in the final product. Furthermore, in addition to energy and raw material savings, the process offers the homogeneous distribution of AIs, high yields, and a relatively short operating time. SSI is usually employed for the AIs with good solubility in SC-CO_2_ and thus poor water solubility. It has been demonstrated that the impregnation of polymer matrices with such AIs successfully improves their dissolution in SBF [123,124,125]. However, more complex systems have also been explored in which a co-solvent is added to the reactor to change the polarity of CO_2_ and the solubility of the hydrophilic drugs in SC-CO_2_ [25,122].

## 3. Drug Delivery from Formulations Prepared by SCF Technologies

The earliest report related to controlled drug delivery dates back to 1952, when a sustained release formulation was first introduced [126]. Since then, various delivery technologies have been developed, which can be divided into three generations. The first generation refers to the development of oral and transdermal controlled release systems. The second generation refers to the development of systems with zero-order release kinetics, self-regulated drug delivery, long-term depot formulations, and the development of nanotechnologies for drug delivery. This also includes research on smart polymers and hydrogels, environmentally sensitive systems (e.g., triggered by *T*, pH), biodegradable systems, etc. The third generation focuses on developing targeted drug delivery (anticancer drugs, siRNA), insulin delivery systems, long-term delivery systems (6–12 months release), and in vitro–in vivo correlation by predispositioning the release profiles. The delivery of AIs with the desired release kinetics requires a sufficient understanding of the physicochemical properties of the active ingredients. In this manner, carrier selection, release mechanisms, and kinetics, which are the most important factors in ensuring an appropriate drug delivery system, can be determined. Before in vivo pharmacokinetic studies are performed, the suitability of the obtained formulations is confirmed by in vitro testing [127]. The United States Pharmacopeia (USP) apparatus is most commonly used for oral and transdermal in vitro drug delivery systems. Seven variations of the USP apparatus (1—basket, 2—paddle, 3—reciprocating cylinder, 4—flow-through cell, 5—paddle over the disc, 6—cylinder, and 7—reciprocating holder) enable an evaluation of drug release from the carrier into the SBF and the determination of release profiles by measuring drug concentrations released over a period of time [128]. Detection is usually performed by UV-Vis spectrophotometry [129,130,131] or high performance liquid chromatography (HPLC) [132,133,134]. In addition to various modifications of the USP apparatus, the oscillating tube apparatus, the Levy–Hayes beaker apparatus, the NF XII apparatus (rotating-bottle type), the Büchner funnel apparatus, and the Wiley apparatus are also employed for drug release testing [135]. In vitro release studies are performed as part of the preliminary testing of drug formulations and serve as quality control to support batch release, to conduct indirect measurements of drug availability, and to predict the impact of formulation methods, drug-carrier interactions, and various other factors on drug bioavailability [128]. Such studies are pivotal for predicting and optimizing drug release kinetics from prepared formulations at lower cost by reducing the number of bioequivalence studies required for scale-up [136].

The formulation of AIs using SCFs enables the development and control of drug particle size [137], as well as the preparation of porous carriers suitable for the loading of AIs [138] and the AI impregnation of carriers [139]. This is feasible by taking advantage of the tunable properties of SCFs, particularly SC-CO_2_, and operating with them under different processing conditions. The interaction of the polymeric carrier with SC-CO_2_ is an important factor in selecting SC technology. Soh et al. [22] emphasized that materials fitting for SAS processing are often not suitable for RESS or PGSS^TM^ as they are not soluble in SC-CO_2_. In addition, PLGA, for example, is promising for SC foaming due to its low glass transition temperature. However, processing PLGA with SAS is reportedly difficult. Furthermore, the choice of SCF technology depends mainly on the interactions of AI with SC-CO_2_. When an AI is nonpolar and highly soluble in SC-CO_2_, the RESS process is primarily used for its micronization. Micronization with RESS greatly increases the dissolution of AIs in SBF compared to untreated AIs, thus significantly improving their bioavailability [140]. SSI can also be employed when AI is soluble in SC-CO_2_ because SC-CO_2_ serves as a transport medium and allows the diffusion of AI into the polymer matrix. Accordingly, nonpolar AIs are impregnated into polymeric carriers, which, due to their (porous) structural properties, allow the enhanced release of AIs into SBF compared to unprocessed AIs [123,124]. On the contrary, when AI is poorly soluble in SC-CO_2_, antisolvent processes (GAS, SAS, etc.) and PGSS^TM^ are used for its micronization or encapsulation. However, depending on the organic solvents and co-solvents used, these methods can also be used for a wider range of AIs of different polarity [19,52,60,141].

Depending on the selected AI, carrier material, and the composition of the final product, the processing of the AI into aerogels can facilitate both the immediate and delayed release of the AI. Many studies have confirmed that embedding the AI in biodegradable polysaccharide aerogels is promising for oral drug delivery. Compared to crystalline AIs, these aerogels achieve more controlled release over a period of several hours to several days of testing [100,102,142]. Compared to the incorporation of AIs into aerogels, SC foaming often achieves long-term release of several days to 1–2 months due to the structural properties of polymer foams (PCL, PLA, PLGA), which is why their application is often focused on biomedical purposes (implant coatings, tissue engineering) [118,143]. The outcomes of AI release studies from SC formulations are summarized in detail in Table 1, which shows the results of AI release according to the SC technology used, the selected AI, and the release system in which the AI is incorporated.

## 4. Conclusions and Outlooks

This paper provides an overview of supercritical fluid technologies as a promising tool for formulating active ingredients, which can be carried out by particle generation or by encapsulating active ingredients in polymeric carriers allowing controlled release. The review outlines the fabrication of drug delivery systems using micronization techniques wherein supercritical CO_2_ can act as a solvent, antisolvent, or solute; the use of supercritical CO_2_ as a drying medium to produce aerogels; supercritical foaming for the fabrication of polymer foams with incorporated active ingredients; and finally, supercritical solvent impregnation for the impregnation of preprepared matrices with supercritical fluid-soluble active ingredients. Demands for investigating novel environmentally friendly processes, solvents, and pharmaceuticals with improved bioavailability are increasing. Compared to conventional methods, supercritical fluid-based processes offer the possibility of environmentally friendly, straightforward, and economical operation (e.g., the supercritical conditions of CO_2_ are achieved at low temperatures) to obtain high-value, solvent-free final products. In addition, depending on the process parameters, the tunable thermodynamic and fluid dynamic properties of supercritical CO_2_ allow for the tailored formulation of active ingredients of different sizes and morphologies, enabling improved drug delivery performance.

Despite the evident advantages and immense progress brought about by supercritical fluid technologies, further research is required to optimize the process parameters and to provide the stability data for the formulations obtained. In addition, methods to improve reproducibility in terms of particle size, the structural characteristics of the products, the yield, and the release kinetics need to be investigated. To date, some supercritical fluid technologies still have not reached industrial implementation due to the lack of in-depth studies on phase behavior, especially in the case of multi-component mixtures (the utilization of co-solvents, multiple active ingredients, etc.). In the future, establishing a detailed database on the impact of process parameters on the final properties of the obtained formulations should be the focus of research for the successful scale-up of supercritical fluid technologies in the pharmaceutical industry. This includes investigating and optimizing controlled and targeted release from supercritical fluid formulations, particularly in developing personalized drug delivery systems that enable the controlled release of active ingredients at concentrations and kinetics tailored to individual needs. The drawbacks of using supercritical fluid technologies in the pharmaceutical industry can also be attributed to the high investment costs and the only recent consideration of environmental concerns, as most legislation continues to allow the use of conventional organic solvents. Once these challenges associated with poor motivation to use environmentally friendly processes and the current knowledge gaps are overcome, supercritical fluid technologies will add a new dimension to the pharmaceutical and biomedical fields regarding the production of novel, sophisticated, and profitable drug delivery systems. Moreover, by investigating supercritical antisolvent fractionation for the fractionation of active ingredients from complex mixtures, the industrial purification of targeted active ingredients with high yields will be possible in the future by means of supercritical fluids.

## Figures and Tables

**Figure 1 pharmaceutics-14-01670-f001:**
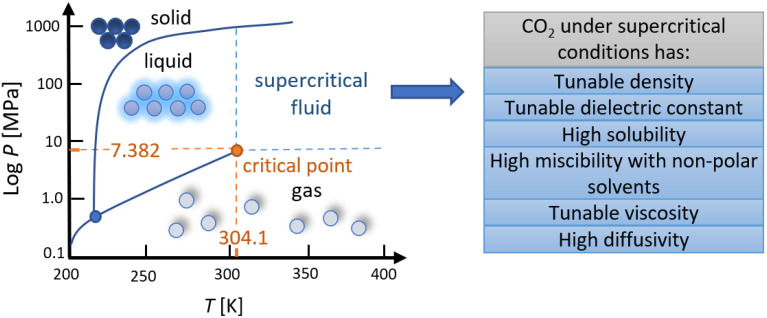
Phase diagram of CO_2_ and its advantageous characteristics at supercritical conditions. Adapted from [22].

**Figure 2 pharmaceutics-14-01670-f002:**
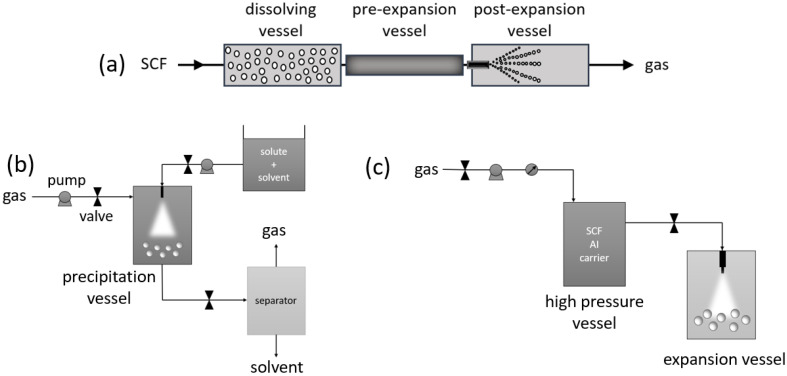
Shematic representation of SCF micronization processes: (**a**) RESS, (**b**) SAS, and (**c**) PGSS^TM^. Adapted from [32,33,34].

**Figure 3 pharmaceutics-14-01670-f003:**
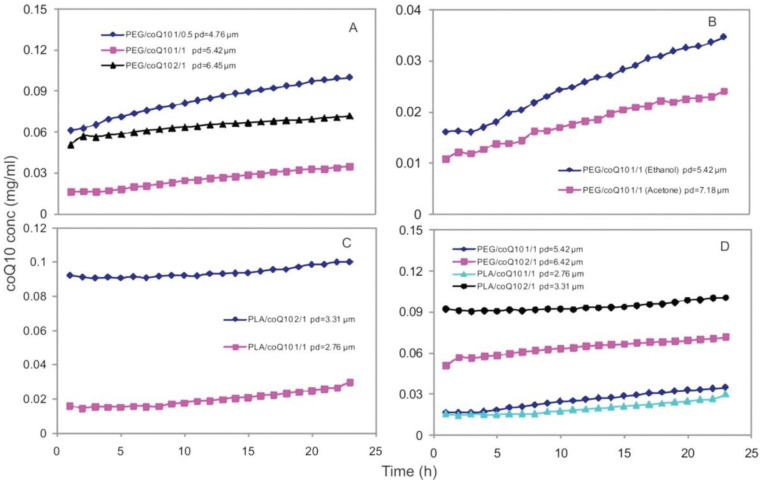
Drug dissolution kinetics, (**A**) the effect of the PEG/coQ10 ratio, (**B**) the effect of the co-solvent, (**C**) the effect of the PLA/coQ10 ratio, and (**D**) a comparison of PEG and PLA [38]. Reprinted with permission from M.d.S. Vergara-Mendoza et al., Microencapsulation of Coenzyme Q10 in Poly(ethylene glycol) and Poly(lactic acid) with Supercritical Carbon Dioxide, Industrial & Engineering Chemistry Research 51(17) 5840–5846, Copyright 2012, American Chemical Society.

**Figure 4 pharmaceutics-14-01670-f004:**
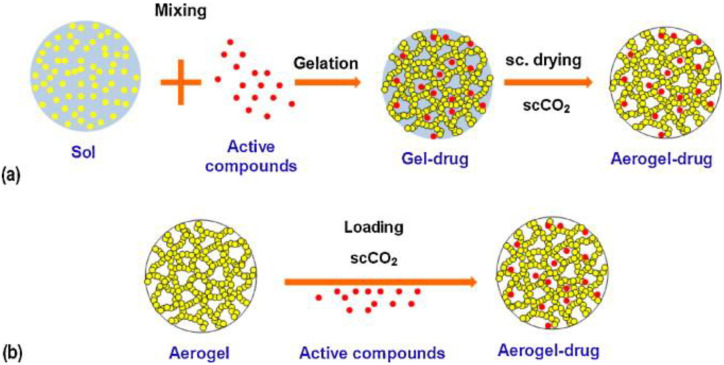
Different strategies for incorporating AIs into aerogels: (**a**) the sol–gel process (co-gelation); (**b**) in the aerogel matrix by supercritical impregnation post-treatment method. Reprinted from Carbohydrate Polymers, 86, García-González et al., Polysaccharide-based aerogels—Promising biodegradable carriers for drug delivery systems, 1425–1438, Copyright (2011), with permission from Elsevier.

**Figure 5 pharmaceutics-14-01670-f005:**
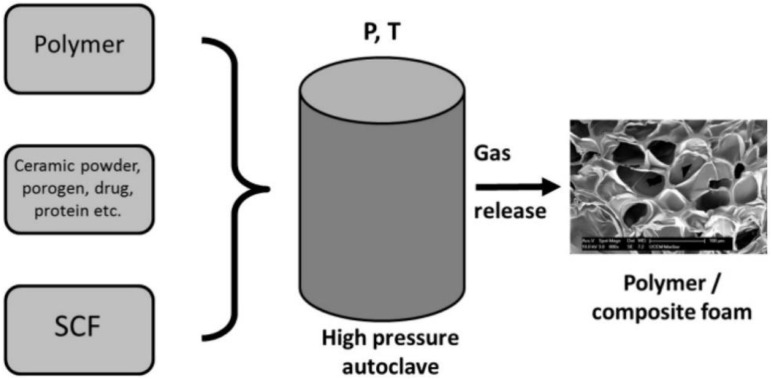
An illustration of the SC foaming leading to stabilized polymer/composite foams [105]. Reprinted from the Energy, 77, Knez et al., industrial applications of supercritical fluids: a review, 235–243, Copyright (2014), with permission from Elsevier.

**Figure 6 pharmaceutics-14-01670-f006:**
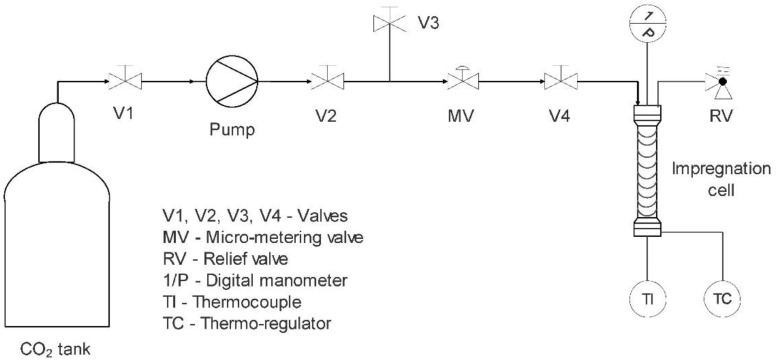
A schematic diagram of SSI apparatus [102]. Reprinted from the Journal of Non-Crystalline Solids, 432, Pantić et al., Supercritical impregnation as a feasible technique for entrapment of fat-soluble vitamins into alginate aerogels, 519–526, Copyright (2016), with permission from Elsevier.

**Table 1 pharmaceutics-14-01670-t001:** The results of AI release in in vitro drug dissolution tests, obtained by the selected SC technique, AI, and system of AI incorporation.

SC Technique	AI	System	AI Release	Reference
RESS	Carbamazepine	AI in SC-CO_2_	Submicron carbamazepine has a dissolution rate coefficient that is up to two times higher than that of the original material.	[144]
	coQ_10_	AI with PEG and PLA in SC-CO_2_ + co-solvent (ethanol/acetone)	PEG: A higher release rate when the concentration of the PEG is higher than that of coQ_10_ (a smaller particle size is produced) and by ethanol as a co-solvent, maintaining the same PEG/coQ_10_ ratio.PLA: a higher release rate when the PLA concentration is higher than coQ_10_.The best dissolution rate occurs at a PLA/coQ_10_ ratio of 2/1.	[38]
	Ethosuximide	AI in SC-CO_2_	Enhanced dissolution rate in PBS compared to the unprocessed material.	[145]
	Fenofibrate	AI in SC-CO_2_	Enhanced dissolution rate in water with 0.05 M SLS: 8.1 times higher dissolution rate coefficient for the micronized AI.	[146]
	Letrozole	AI in SC-CO_2_ + co-solvent (menthol)	Improved dissolution rate: 14.86 times higher dissolution rate coefficient for the micronized drug.	[147]
	Lonidamine	AI in SC-CO_2_	Improved dissolution rate of the micronized drug in aqueous media.	[148]
	Naproxen	AI in SC-CO_2_	Improved dissolution rate in SBF: higher dissolution rate coefficients of the micronized drug compared to the unprocessed drug at pH = 2.0 and pH = 7.4.	[149]
	Progesterone	AI in SC-CO_2_	Enhanced drug dissolution rate after RESS treatment.	[150]
SAS	Cetirizine dihydrochloride and ketotifen	AIs in zein and SC-CO_2_	Prolonged (controlled) release of both processed antihistamines.	[151]
	Curcumin	AI and poly (vinyl pyrrolidone) in an ethanol/acetone mixture with SC-CO_2_	Up to 600 times increased solubility of the processed AI compared to unprocessed.	[45]
	Fenofibrate	AI in the polymers P407 and TPGS with SC-CO_2_	95.1% ± 2.5% improved drug dissolution rate compared to the unprocessed drug.	[19]
	Ketoprofen and nimesulide	AIs in β-cyclodextrin with SC-CO_2_	An enhancement of the drug dissolution rate of up to 21 (nimesulide) and 7 (ketoprofen) times.	[152]
	Mangiferin	AI with N, N-dimethylformamide (DMF) as the solvent and SC-CO_2_ as the antisolvent	4.26, 2.1, and 2.5 times better solubility of the processed AI in water, simulated gastric fluid, and simulated intestinal fluid, respectively.	[153]
	Rutin	AI in acetone and DMSO with SC-CO_2_	A dissolution rate of micronized AI particles up to 10 times faster than nonprocessed AI.	[141]
	Trans-resveratrol	AI in alcohol (methanol or ethanol) and dichloromethane mixtures with SC-CO_2_	Improved release rate of the processed drug.	[154]
GAS	Rosemary extract	AI encapsulated in PCL dissolved in dichloromethane, antisolvent SC-CO_2_	Burst release in an aqueous medium, first-order kinetic model.	[155]
PGSS^TM^	Epigallocatechin gallate	AI in OSA-starch, soybean lecithin and β-glucan with SC-CO_2_	Rapid release for polysaccharide matrices, namely OSA-starch and β-glucan, and somewhat more controlled release for amphiphilic lecithin.	[60]
	Eucalyptol	PEG and/or PCL with SC-CO_2_	Significantly delayed release of AI in PEG and/or PCL compared to the pure AI (an average of 40% released AI from the polymer and 96% released unencapsulated AI in 120 min).	[56]
	Fenofibrate	AI in Gelucire^®^ 50/13 with SC-CO_2_	Slow, controlled release	[52]
	Fenofibrate, nimodipine and o-vanillin	AIs in Brij S100 and PEG 4000 with SC-CO_2_	Increased dissolution rate of Brij S100 micronized nimodipine, Brij S100 micronized fenofibrate, and Brij S100/PEG 4000 micronized o-vanillin compared to the unprocessed AIs.	[59]
	Ibuprofen	AI in pluronic poloxamers, gelucire and glyceryl monostearate with SC-CO_2_	Accelerated release rate of AI in pluronic carriers, prolonged/controlled release in gelucire and glyceryl monostearate.	[54]
	Nifedipine	AI in PEG 4000 with SC-CO_2_	Increased dissolution rate of micronized AI compared to the unprocessed AIs.	[156]
	Omega-3 polyunsaturated fatty acids and astaxanthin-rich salmon oil	AIs in PAG-6000 with SC-CO_2_	Rapid release of oil in distilled water: up to 65% within 30 min.	[55]
Aerogels	Ampicillin	AI loaded liposomes entrapped in alginate aerogels	Slow and controlled release of AI from aerogel over 100 h compared to the burst release of pure AI within the first 5 h.	[96]
	Celecoxib	AI in potato starch aerogel	Faster dissolution rate of AI from aerogel compared to pure AI in simulated gastric and intestinal fluids over a period of 7 h. The release kinetics follow the Korsmeyer–Peppas model.	[94]
	Curcumin	AI in pectin- and chitosan-coated pectin aerogels	Enhanced dissolution of AI from aerogels after 2 h in gastric fluid and 22 h in intestinal fluid. The fastest AI release is obtained from pure pectin aerogels.	[97]
	Diclofenac sodium, indomethacin	AIs in pectin and xanthan aerogels	Release of the two AIs within 24 h. The release profile of indomethacin showed a higher initial release rate compared with diclofenac and slower release after 5 h of testing.	[70]
	Esomeprazole	AI incorporated in alginate, pectin, chitosan, and composite aerogels via diffusion or supercritical impregnation	Slower and more controlled release of AI from aerogels in gastric and intestinal fluids compared to pure AI. The slowest drug release is achieved from pectin and chitosan composite aerogels.	[100]
	Ibuprofen, ketoprofen, triflusal	AIs in 14 silica-gelatin aerogels of different composition	Depending on the composition of the aerogels, both immediate and delayed release are possible.	[142]
	Ketoprofen, quercetin	AIs loaded in pure alginate and composite pectin, κ-carrageenan, and alginate aerogel microparticles by supercritical impregnation	The release of AIs from the aerogels is slower and more controlled than that of unprocessed AIs within the 60-min test period.	[101]
	Nifedipine	Guar, xanthan, pectin, and alginate aerogels prepared by novel ethanol induced gelation	Prolonged release of AI up to 14 days for guar and xanthan aerogels, steady release within 6 h for alginate and pectin aerogels in simulated gastric and intestinal fluids. Drug release from pectin aerogels is controlled by the Hixson-Crowell model, from alginate in PBS by the first-order model, and in HCl media by the Korsmeyer–Peppas model.	[82]
	Resveratrol	AI loaded in TEMPO-oxidized cellulose aerogels	After the initial burst release (within the first 15 min), controlled release of AI from aerogels in simulated gastric and intestinal fluids is achieved. After 5 h, 35–50% of the AI is released from the aerogels, compared to 90% of pure AI within the same time period.	[93]
	Theophylline	AI loaded in pectin aerogels prepared with different solution pH and calcium concentrations	Drug release from all aerogels shows an initial burst release followed by a more controlled release. The low pH of the pectin starting solutions results in faster release of the AI, while calcium crosslinking decreases the rate of AI release. The main release mechanism is shown to be the Peppas-Sahlin model.	[98]
	Tetracycline hydrochloride	κ-carrageenan aerogels prepared with the addition of potassium salts as crosslinking agents	Initial burst release followed by a plateau at approximately 60 min, corresponding to 90% of the released active ingredient in PBS, with a pH of 7.4.	[73]
	Vancomycin	AI loaded in chitosan aerogel beads	Burst release within the first hour, followed by a plateau during the remaining test period (2 days). The release profile is fitted to a first-order release model.	[95]
	Vitamin D_3_	AI loaded in alginate aerogels	Significantly improved dissolution of impregnated AI compared to crystalline AI within 7 h.	[102]
SC foaming	Cinnamaldehyde	AI in PLA foam	An initial burst release, followed by a slowed release over the 300-min test period; Quasi-Fickian diffusion, fitting the Korsmeyer–Peppas mathematical model.	[117]
	Curcumin, gentamicin	AIs loaded in PLGA foam	Diffusion-controlled release; the drugs were not completely released in the 14-day test period. A slower release is obtained for curcumin.	[112]
	Gemcitabine	AI in PLGA foam	An initial burst release with over 80% of the drug released in the first 5 days, followed by prolonged release over the 20-day test period. The drug release is first controlled by a diffusion process, followed by the internal transfer of mass and polymer degradation.	[116]
	DNA	DNA loaded in PLGA or composite chitosan/PLGA foams	An initial burst release followed by slow release over a 40-day period for composite foams.	[111]
	Mesoglycan	AI loaded in PCL foam	Dissolution tests demonstrated prolonged release of the AI from the PCL foam of up to 70 times longer compared to the pure AI during the 3-day testing period.	[143]
	Nimesulide	AI in PCL foam	3.5 times prolonged release of the AI from the PCL foam compared to the pure AI in the 3-day test period.	[115]
	Paclitaxel	AI in PLGA or PLGA-PEG foams	Continuous and nearly linear AI release from the foams, with approximately 50% of release within 8 weeks.	[118]
	Thymol	AI loaded in PLA and PLGA foams	Prolonged release of the AI in PBS over the 1.5-month testing period.	[110]
	Transglutaminase	AI crosslinked with glutaraldehyde in PCL foam containing chitosan and hydroxyapatite	Prolonged AI release for up to 30 days.	[119]
SSI	Acetylsalicylic acid	AI in barley and yeast β-glucan aerogels	Faster release of AI from barley aerogel and more sustained release from yeast β-glucan aerogel during the 25-h test period.	[86]
	Cholesterol	AI loaded in PMMA, PMMA/PCL microspheres	Faster dissolution of the AI in PMMA and more sustained release from PMMA/PCL during the 450-h test period.	[124]
	Fenofibrate	AI in mesoporous silica	Improved drug dissolution of impregnated AI compared to crystalline AI during the 120-min test period.	[125]
	Flurbiprofen	AI in PMMA/β-tricalcium phosphate biocomposites	50% of the AI released within the first 4 h of measurement in an ethanol solution.	[157]
	Ibuprofen	AI in Soluplus^®^	Improved dissolution of AI loaded by SSI compared to the physical mixture during the 140-h test period.	[158]
	Ketoconazole	AI in poly (vinyl pyrrolidone) (PVP) and hydroxy propyl methyl cellulose (HPMC)	Improved dissolution of AI impregnated in polymers by SSI compared to the physical mixture during the 75-min test period.	[159]
	Ketoprofen	AI in PVP	Fastest release of AI (87% in the first 30 min) from the impregnated polymer compared to the physical mixture with crystalline or amorphous AI (micro-tablets). Drug dissolution is controlled by polymer degradation.	[123]

## Data Availability

Not applicable.

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
