# Peer review of "Supercritical Fluid Technologies for the Incorporation of Synthetic and Natural Active Compounds into Materials for Drug Formulation and Delivery"

_pharmaceutics, 2022, doi:10.3390/pharmaceutics14081670_

Round 1

Reviewer 1 Report

The manuscript “ Supercritical Fluid Technologies for the Incorporation of Synthetic and Natural Active Compounds into Functional Materials”  is publishable in the Pharmaceutic journal in its current form.

The manuscript is dealing with a review work on the application of the supercritical fluid in the field of pharmaceutic. The use of supercritical fluid has been discussed to controlled release of the drug in body media. The review covers the successful fabrication and application of the same  for the processing and incorporation of active compounds. 

Author Response

REVIEWER REPORT(S): Referee: 1 The manuscript “Supercritical Fluid Technologies for the Incorporation of Synthetic and Natural Active Compounds into Functional Materials” is publishable in the Pharmaceutic journal in its current form. The manuscript is dealing with a review work on the application of the supercritical fluid in the field of pharmaceutic. The use of supercritical fluid has been discussed to controlled release of the drug in body media. The review covers the successful fabrication and application of the same for the processing and incorporation of active compounds. Answer: We thank the reviewer for the positive feedback on our work and the recommendation for publication. __________________________________________________________________________________

Reviewer 2 Report

This manuscript summarizes applications of supercritical fluid (SCF) technology in drug formulation and delivery. Overall this is a nice addition to the literature, but the scope of this manuscript is not clear, and the presentation needs to be refined.

1. The title mentions "synthetic and natural active compounds", and the abstract starts with "polyphenolic compounds" and "synthetic drugs". A significant portion of the abstract is also devoted to polyphenol compounds.
It is not clear whether the authors want to focus on "polyphenolic compounds" or "natural active compounds" in general. It appears that other natural compounds, such as DNA, mesoglycan and Thymol, are included in Table 1. And there is no discussion on why SCF is specifically suited for polyphenols as compared to other natural active compounds.

The title also says "functional materials", while the manuscript only focuses on the materials for drug formulation and delivery.

So the title, abstract and introduction should be revised to more accurately represent the scope of the manuscript.

2. This manuscript does not have any original figures. It's ok to include some images and plots from other papers, but the schemes such as those in Figure 1, 2, and 4 are the most creative part of a review paper, and should be original if possible.

3. Some minor issues in figures.
The styles for panel labels are not consistent (upper vs lower cases, parentheses)
Figure 2: panel (a) has "SCF" while panels (b) and (c) have "CO2", which are not consistent.
Figure 4 (line 306). The figure label is missing and the caption is not complete.

4. Line 109-110: "
When smaller particles are used, their solubility increases due to their 109 higher surface area in contact with water, therefore requiring lower drug dosages [28]." This statement is not accurate. According to Ref. 28, the increased solubility is due to not only the higher specific surface area, but also crystal lattice defects and a change in surface properties.

Author Response

Referee: 2

This manuscript summarizes applications of supercritical fluid (SCF) technology in drug formulation and delivery. Overall this is a nice addition to the literature, but the scope of this manuscript is not clear, and the presentation needs to be refined.

Answer: We thank the reviewer for her/his thorough review of our work and the points highlighted. We have taken into account all the suggestions. In addition, the valid issues raised by the reviewer have contributed to the (now) increased quality of the revised article and are thus beneficial to the Pharmaceutics readership.

  1. The title mentions "synthetic and natural active compounds", and the abstract starts with "polyphenolic compounds" and "synthetic drugs". A significant portion of the abstract is also devoted to polyphenol compounds.

It is not clear whether the authors want to focus on "polyphenolic compounds" or "natural active compounds" in general. It appears that other natural compounds, such as DNA, mesoglycan and Thymol, are included in Table 1. And there is no discussion on why SCF is specifically suited for polyphenols as compared to other natural active compounds.

The title also says "functional materials", while the manuscript only focuses on the materials for drug formulation and delivery.

So the title, abstract and introduction should be revised to more accurately represent the scope of the manuscript.

Answer: We appreciate the comment. As suggested, the term polyphenolic compounds in the abstract has been eliminated and replaced with the more general term active compounds because the review includes other natural compounds in Table 1.

The title has been changed to "Supercritical fluid technologies for the incorporation of synthetic and natural active compounds into materials for drug formulation and delivery" as suggested.

  1. This manuscript does not have any original figures. It's ok to include some images and plots from other papers, but the schemes such as those in Figure 1, 2, and 4 are the most creative part of a review paper, and should be original if possible.

Answer: We thank the reviewer for her/his comment. Although it might appear that these figures were copied from other sources, we would like to mention that Figures 1 and 2 are original works of the authors.

Figure 1 depicts phase diagrams of CO2 and Figure 2 depicts various micronization processes, both of which are generally known and can be found in other papers that are cited in the figure captions.
For Figure 4, we have received copyright permission, and the citation is provided in the figure caption. We would like to kindly ask the reviewer to accept our way of presenting figures.

  1. Some minor issues in figures.

The styles for panel labels are not consistent (upper vs lower cases, parentheses)

Figure 2: panel (a) has "SCF" while panels (b) and (c) have "CO2", which are not consistent.

Figure 4 (line 306). The figure label is missing and the caption is not complete.

Answer: We are thankful for the remark. The problem with the inconsistent styles for panel labels must have occurred when MDPI converted our original Word document to an MDPI template. We have now addressed the issue and corrected the styles in a revised document converted to an MDPI template.

The new Figure 2 is now consistent and contains a more generalized term “gas” instead of CO2. We have additionally provided the figure label (caption) for Figure 4.

  1. Line 109-110: "When smaller particles are used, their solubility increases due to their 109 higher surface area in contact with water, therefore requiring lower drug dosages [28]." This statement is not accurate. According to Ref. 28, the increased solubility is due to not only the higher specific surface area, but also crystal lattice defects and a change in surface properties.

Answer: We are grateful for this observation. As suggested by the reviewer, we have extended our statement and included the influence of crystal lattice defects and a change in surface properties on a higher solubility.

Reviewer 3 Report

This manuscript is offering a good overview of recent advances in the field of supercritical fluid technologies.

All techniques are well described with relevant examples from the literature. 

Author Response

This manuscript is offering a good overview of recent advances in the field of supercritical fluid technologies.

All techniques are well described with relevant examples from the literature.

 Answer:  Thank you for the positive feedback. We are pleased to hear that the original manuscript contains a comprehensive overview of recent advances in the field of SC technologies with relevant examples, which was our main aim.

Round 2

Reviewer 2 Report

The authors have addressed all my concerns.